# Long-Term Chromatic Durability of White Spot Lesions through Employment of Infiltration Resin Treatment

**DOI:** 10.3390/medicina59040749

**Published:** 2023-04-12

**Authors:** Francesco Puleio, Federica Di Spirito, Giuseppe Lo Giudice, Giuseppe Pantaleo, David Rizzo, Roberto Lo Giudice

**Affiliations:** 1Department of Biomedical and Dental Sciences and Morphofunctional Imaging, Messina University, 98100 Messina, Italy; francesco.puleio@unime.it; 2Department of Medicine, Surgery and Dentistry “Schola Medica Salernitana”, University of Salerno, 84084 Baronissi, Italy; fdispirito@unisa.it; 3Department of Clinical and Experimental Medicine, Messina University, 98100 Messina, Italy; logiudiceg@unime.it (G.L.G.); roberto.logiudice@unime.it (R.L.G.); 4Independent Researcher, 98124 Messina, Italy; dr.davidrizzo@libero.it

**Keywords:** white spot lesions, enamel demineralization, infiltration technique, resin infiltration

## Abstract

*Background and Objectives*: White spot lesions (WSLs) denote regions of subsurface demineralization on the enamel that manifest as opaque and milky-white regions. Treatment for WSLs is essential for both clinical and aesthetic reasons. Resin infiltration has been identified as the most efficacious solution for alleviating WSLs, but studies with long-term monitoring are scarce. The aim of this clinical study is to assess the color change stability of the lesion after four years of implementing the resin infiltration technique. *Materials and Methods:* Forty non-cavity and unrestored white spot lesions (WSLs) were treated with the resin infiltration technique. The color of the WSLs and adjacent healthy enamel (SAE) was assessed using a spectrophotometer at T0 (baseline), T1 (after treatment), T2 (1 year after) and T3 (4 years after). The Wilcoxon test was utilized to determine the significance of the variation of color (ΔE) between WSLs and SAE over the observed time periods. *Results*: When comparing the color difference ΔE (WSLs-SAE) at T0-T1, the Wilcoxon test demonstarated a statistically significant difference (*p* < 0.05). For ΔE (WSLs-SAE) at T1-T2 and T1-T3, the color variation was not statistically significant (*p* = 0.305 and *p* = 0.337). *Conclusions:* The study’s findings indicate that the resin infiltration technique is an effective solution for resolving the appearance of WSLs, and the results have demonstrated stability for a minimum of four years.

## 1. Introduction

White spot lesions (WSLs) are defined as the subsurface demineralization of enamel [1]. This form of demineralization modifies the refractive index of the enamel, and the contrast in refractive index between the healthy enamel and the demineralized area generates a lesion that exhibits a milky-white opaque appearance, which is easily distinguishable from the neighboring healthy enamel [2].

The etiology of WSLs is diverse, as they may stem from systemic causes, such as excessive fluoride intake during childhood, or they may be acquired later, for example, due to trauma, excessive etching of the labial enamel surface during placement of orthodontic-fixed appliances, or inadequate oral hygiene, particularly in patients undergoing fixed orthodontic therapy [3,4]. According to Tufekci et al., 46% of patients undergoing fixed orthodontic therapy develop at least one lesion [5]. Over the past few decades, the prevalence of enamel WSLs has risen due to an increased susceptibility to enamel demineralization during orthodontic treatment, as the appliances create challenges for oral hygiene. Additionally, iatrogenic WSLs are caused by excessive surplus etching of the labial enamel surface during placement of orthodontic-fixed appliances. As per reports, the incidence of postorthodontic WSLs can be as high as 96% [5].

Hence, it is recommended to diagnose and document these lesions using standardized photographic plates, taking into account magnification, exposure time, lighting, etc., before initiating orthodontic treatment. WSLs identified prior to orthodontic treatment are considered a risk factor for the development of new lesions, with poor oral hygiene, excessive consumption of fermentable carbohydrates, prolonged etching time, treated/decayed molars, excess bonding, frequent drinking, and treatment duration being other potential risk factors [6].

High bacterial concentrations reduce plaque to a greater extent in orthodontic patients than in others, leading to faster progression of caries in patients with a full set of orthodontic appliances. A significant increase in the prevalence of these lesions has been reported around the bases of brackets or between brackets/bands, and in the gingival margins in the cervical areas and middle thirds of teeth under orthodontic wires, as well as with full-coverage rapid maxillary expanders [7].

During the first six months of formation, WSLs may undergo remineralization; however, this process is slow and occurs only in the outermost 30 microns of the enamel, resulting in incomplete resolution of the lesions and leaving them visible to the naked eye [6,7]. Most WSLs maintain their shape with stable mineralization over time [8]. About 15% of these lesions tend to worsen, requiring more invasive restorative therapies [8]. Therefore, treatment of these lesions is essential not only for the resolution of the aesthetic problem but also for avoiding the worsening of the clinical situation.

WSLs should be managed using a multifactorial approach, with the most important strategy being the prevention of demineralization and biofilm formation. Prevention should begin by educating and motivating the patient to comply with a noncariogenic diet and maintain oral hygiene. Effective oral hygiene is the cornerstone of prophylactic measures in fixed-orthodontic patients [3]. Mechanical plaque control and removal by proper brushing of the tooth surfaces, at least twice daily, with fluoride-containing toothpaste, especially in biofilm retention areas, is strongly recommended. During recall visits, patient motivation should be reevaluated and, if necessary, oral hygiene and dietary instructions should be repeated, and tooth surfaces should receive professional cleaning [5,6]. 

Professional prophylactic cleaning reduces bacterial load, increases the efficacy of brushing, and facilitates cleaning by the patient. Professional tooth cleaning two or three times a year helps maintain a healthy mouth, decreasing the risk of dental caries and the number of teeth with carious lesions. Fluoridated pastes with progressively finer particle sizes can be used to polish coronal surfaces. Furthermore, elastomeric polishing cups or brushes help prevent mechanical retention of bacteria. Along with brushing frequency, patient age, time elapsed since appliance removal, length of treatment, the type of tooth (central or lateral incisor) and the surface area of the WSLs also have an effect on WSL improvement, as mentioned in refs. [2,8].

Some therapies suggested in the literature propose remineralization of the lesion through the use of agents containing 5% fluoride or casein phosphopeptide (CPP-ACP). However, due to their inability to penetrate the complete thickness of the lesion, these agents do not provide a solution to the aesthetic concern, as referenced in sources [9,10,11].

Fluoride has been shown to play a favorable role in preventing WSLs through the use of various methods, including fluoride mouthwashes, fluoride varnishes, fluoride gels, fluoride in bonding agents, fluoride toothpastes and fluoride in elastomers. This is due to the fluoride ion’s ability to modify bacterial metabolism in dental plaque by inhibiting enzymatic processes and the production of acids, altering the composition of bacterial flora and/or the metabolic activity of microorganisms, and promoting remineralization of carious lesions at early stages through a remineralization effect, particularly at low concentrations. This information is supported by references [8,10].

Other therapies, such as microabrasion, involve the use of 6.6% hydrochloric acid and 20-to-160 μm sized silicon carbide microparticles to remove a superficial layer of enamel. Due to its relatively invasive nature, it was believed that delayed application was beneficial, given improvements of lesions through saliva-based remineralization and spontaneous surface abrasion subsequent to debonding. While this is a useful method for the treatment of postorthodontic WSLs, the depth of the lesion should be under 0.2 mm and it might be associated with the bleaching technique. However, even this technique does not seem effective for solving the aesthetic problem, as referenced in sources [12,13].

The most effective technique for resolving the lesion is resin infiltration, as proven by reference [14]. This involves an initial etching phase with 15% hydrochloric acid (HCL). Subsequently, the dental surface is dried with 99% alcohol, and finally infiltrated with a resin that is light-cured. This resin has a refractive index like that of enamel, masking the opaque white appearance typical of WSLs, as mentioned in reference [15].

The benefits of resin infiltration are numerous. Firstly, it is the most effective and predictable treatment for the aesthetic resolution of WSLs when compared with remineralizing techniques or microabrasion, as referenced in source [14]. Secondly, it is able to penetrate the lesion deeper than remineralization techniques, thanks to etching with HCL, as mentioned in sources [16,17,18,19]. Thirdly, the absence of gaps after the resin infiltration inhibits bacterial proliferation and WSL progression, as supported by reference [20]. Additionally, the dental surface is shinier than remineralization treatments, resulting in a lower deposit of bacterial plaque, as referenced in source [21]. However, there are no studies in the literature with long follow-ups of teeth with WSLs treated with this technique. The objective of this clinical study is to evaluate the stability of the color change of the lesion after 4 years of using the resin infiltration technique.

## 2. Materials and Methods

The study design was an in vivo, single-center, experimental, prospective study. All patients were fully informed and signed an informed consent form to be included in the study. The research was conducted in accordance with the Declaration of Helsinki, and the protocol was approved by the Ethics Committee of 19-2018 of 23 April 2018.

The study included 40 non-cavity, unrestored, idiopathic or postorthodontic white spot lesions (WSLs) that were treated with the resin infiltration technique (Icon, DMG, Hamburg, Germany). All lesions were treated by the same operator. The subjects were recruited with the exclusion criteria of dental diseases, including hypocalcification, hypoplasia, fluorosis, and hypoplastic molar-incisive syndrome. The sample size was determined with a type-I error of 0.05 and a power of 0.85, using the parameter ΔE WSL vs. sound adjacent enamel (SAE) after treatment.

The treatment of the white spots was carried out according to the operational steps suggested by the manufacturer (ICON, DMG, Hamburg, Germany), which included the following:Isolation of the treated tooth with a rubber dam;Removal of superficial plaque;Etching of the lesion surface with 15% hydrochloric acid (HCL) for 120 s;Washing with air and water for 30 s;Application of 99% ethanol dehydrating solution for 30 s;Repeating of the etching process for 2 min if the WSL is still visible (for a maximum of three times);Drying of the lesion with air;Application of infiltrating resin for 30 s;Elimination of excess with air;Light curing for 40 s.

Observations were performed before and after treatment (T0 and T1, respectively), after 12 months (T2), and 48 months (T3). The measurements were performed by a second investigator who was blinded to the treatment used. The color of the WSLs and adjacent healthy enamel (SAE) was evaluated using a spectrophotometer (Spectroshade Micro Device; Medical High Technologies, Verona, Italy), using the CIE L*a*b* system. (Figure 1)

The color change was calculated using the following formula [22,23]:ΔE (Par 1 − Par 2) = [(LPar1 − LPar2)^2^ + (α Par1 − α Par2)^2^ + (β Par1 − β Par2)^2^]^1/2^

The color of the lesion and of the SAE was measured at 3 random points, and the mean value was then calculated [22]. Additionally, a photographic check was conducted, and the spectrophotometer images were analyzed by a third operator. The difference of ΔE values between WSLs and SAE was calculated for each observation period (T0-T1, T1-T2 and T1-T3), which was considered as a control reference. The Wilcoxon test was used to evaluate the significance of the variation of ΔE between WSLs and SAE over the observed time periods. The data analysis was performed using the SPSS software for Windows ver. 22 (IBM, Milan, Italy).

## 3. Results

After a period of four years, all patients (100%) were evaluated with no loss of data. Table 1 describes the ΔE values recorded for WSLs vs. SAE at four observation times.

The Wilcoxon test revealed a statistically significant difference (*p* < 0.05) in the color difference ΔE (WSLs-SAE) between T0-T1. However, there was no statistically significant color variation for ΔE (WSLs-SAE) between T1-T2 and T1-T3 (*p* = 0.305 and *p* = 0.337, respectively). Out of the four treated patients, a significant color change was observed between T0 and T1, but the color was not stable between T1 and T2. These results suggest that an altered enamel morphology may have influenced the tooth porosity and, consequently, the tooth color.

The significance of the Wilcoxon test at T0-T1 shows that the technique is effective in determining a color change. The non-significance of the Wilcoxon test at T1-T2 and T1-T3 explains instead that, after applying the technique and changing the color of the lesion, there was no color change after 12 and 48 months; the color is kept stable.

White spot lesions (WSLs) are not only a cosmetic concern but also an indication of early-stage tooth decay, which may require restorative treatment if left untreated (refs. [24,25,26]). The resin infiltration technique has been extensively studied and found to be effective in resolving WSLs (refs. [15,27,28,29,30,31]). This technique involves etching the lesion with 15% HCL for 2 min to make it porous and receptive to the resin, followed by dehydrating the lesion with 99% alcohol. Additional etching steps may be required if the whitish-opaque appearance persists. The resin is then applied and cured with light. Due to its high infiltration coefficient and refractive index similar to healthy enamel, the resin is able to mask the opaque white appearance of the lesion. Moreover, the resin prevents the progression of the lesion and tooth decay, while improving the hardness and acid resistance of the dental tissue (refs. [32,33]).

However, there are no published studies that have evaluated the long-term stability of the aesthetic results obtained from this technique in a clinical setting. Further research is needed to examine the effects of the chemical composition of the resin used for treatment, particularly TEGDMA, which has a higher water absorption rate compared to other resins like BisGMA and UDMA [34,35]. This property has been linked to an increased risk of pigmentation due to water’s ability to transport pigments [36,37]. Our research aimed to evaluate the long-term stability of the resin infiltration technique for treating WSLs by using a previously established methodology [22]. The data we collected showed a statistically significant difference in color change (ΔE) between WSLs and surrounding enamel at T0 and T1, but no significant difference at later time points. This suggests that the technique is effective in improving the appearance of WSLs and that the results remain stable for at least 4 years. However, in 2 out of 40 cases (5%), the technique was not effective. The literature suggests that deep and smooth lesions may not be fully resolved by resin infiltration [23,38]. It is possible that these two lesions that did not register a color change after infiltration of the resin were too deep to be infiltrated by the resin itself: the application of HCL 15% for 2 min determines enamel substance loss of 34 μ m approximately, but 29% of WLSs have a surface layer thicker than 50 μm [39,40]. However, the depth of the lesion was not measured in any way before performing the technique, therefore the authors cannot give an answer as to why these two lesions were not resolved. Not all types of WSLs can therefore be treated with resin infiltration for the resolution of the aesthetic problem. Additionally, in 1 out of 40 cases, the lesion improved at T1 but reappeared at T2. There is currently no standard method for predicting the outcome or stability of the resin infiltration technique before it is applied [41]. The current research has limitations due to the limited number of participants examined. Additional investigation in this area should consider the progression of these lesions over a longer period of time.

## 4. Conclusions

Enamel decalcification surrounding orthodontic appliances is a common issue during and after orthodontic treatment. Patients can manage these lesions by practicing good oral hygiene and receiving prophylaxis with topical fluoride. Other methods, such as tooth bleaching, microabrasion, and antiseptics have also been recommended. The resin infiltration technique has been shown to effectively resolve white spot lesions (WSLs), which not only pose a cosmetic concern but may also indicate an early stage of tooth decay. Our study found that the technique was effective in most cases, but 5% of cases did not respond to the treatment and one case experienced lesion improvement followed by reappearance. Further research is needed to assess the long-term stability of the results and the effects of the resin’s chemical composition. The study’s limitations include a small sample size, and additional investigation is necessary to evaluate the progression of these lesions over an extended period.

## Figures and Tables

**Figure 1 medicina-59-00749-f001:**
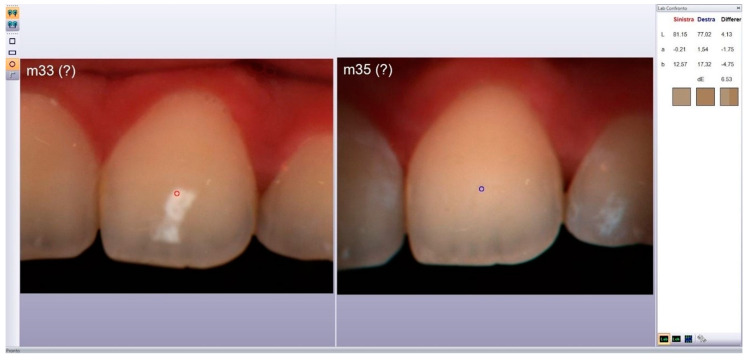
MHT SpectroShade software comparing pictures using CIE L*a*b* system.

**Table 1 medicina-59-00749-t001:** ΔE (WSL vs. SAE) at T0, T1, T2, and T3 for each patient.

Pz.		ΔE WSL vs. SAE			Pz.		ΔE WSL vs. SAE		
	T0	T1	T2	T3		T0	T1	T2	T3
1	10.57	5.63	6.48	5.52	21	10.25	5.18	5.87	4.98
2	11.5	3.18	2.57	2.93	22	12.218	4.87	5.94	4.35
3	10.25	3.67	3.57	4.94	23	11.053	5.66	5.82	4.59
4	5.98	2.8	2.95	2.38	24	10.25	5.33	6.82	5.72
5	12.46	5.38	6.84	5.39	25	10.25	5.92	4.76	5.76
6	11.82	5.67	5.74	6.34	26	11.053	4.16	4.98	5.23
7	12.87	11.64	12.22	13.35	27	5.98	2.76	3.23	4.22
8	9.48	3.58	3.57	4.82	28	10.25	4.86	3.78	3.59
9	6.65	3.83	4.53	3.38	29	10.25	5.99	4.65	3.12
10	11.46	5.34	5.74	5.37	30	10.25	4.74	5.12	4.22
11	11.89	4.71	4.12	5.37	31	11.053	3.54	5.23	5.87
12	12.45	5.93	6.23	7.11	32	12.218	11.32	11.95	12.7
13	12.19	5.73	6.32	5.93	33	11.053	4.86	5.23	6.88
14	12.218	4.96	4.65	3.76	34	10.25	5.56	4.26	3.32
15	11.053	4.69	3.87	4.85	35	12.218	4.15	4.28	5.66
16	7.54	4.95	3.74	3.74	36	12.218	4.17	5.87	4.54
17	12.218	6.63	7.45	7.93	37	12.218	6.16	5.83	4.29
18	12.218	6.4	7.98	6.35	38	10.25	5.82	6.12	5.02
19	8.59	4.96	3.41	4.36	39	10.25	4.84	9.28	10.1
20	10.57	3.47	2.29	3.83	40	11.053	6.38	5.24	6.32

## Data Availability

The data presented in this study are available on request from the corresponding author.

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
