# Peer review of "Long-Term Chromatic Durability of White Spot Lesions through Employment of Infiltration Resin Treatment"

_medicina, 2023, doi:10.3390/medicina59040749_

Round 1

Reviewer 1 Report

In the article entitled "Long-Term Color Stability of White Spot Lesions Using Infiltration Resin Treatment," the authors tried just to increase the follow-up time of one of the current treatment approaches. Although I do believe that the novelty of this work is low, it can provide some useful information for readers. However, before the final decision, English proficiency should be elevated, and the lost details should be added. For example, on page 3, line 6, light curing for 40_it should be 40 S.

Author Response

In the article entitled "Long-Term Color Stability of White Spot Lesions Using Infiltration Resin Treatment," the authors tried just to increase the follow-up time of one of the current treatment approaches. Although I do believe that the novelty of this work is low, it can provide some useful information for readers. However, before the final decision, English proficiency should be elevated, and the lost details should be added. For example, on page 3, line 6, light curing for 40_it should be 40 S.

The English review and implementation was done by a qualified center and the certification was sent to the editor.

The 40 second change was made in the text

Reviewer 2 Report

Dear authors,
the work is interesting, albeit not particularly innovative, as it proposes a medium-term follow-up, however I believe some aspects of it should be improved:

-The name list in the manuscript must be adjusted according to your submission data.

-In the material section authors should specify the setting and how the patient were recruited, the sample selection represents an important source of bias; nothing is specified about the initial characteristics of the WLS of the patients enrolled in the study.

-How was possible to blind the observer if there is just one treatment option evaluated? Considering that to evaluate the effectiveness of the treatment it is necessary to compare the treated tooth with the adjacent one?

-How was the photographic check mentioned in the study carried out?

-The results should be reported in more detail, explaining their significance;

-The discussion must be improved, the first part is redundant.

overall I think the work could be improved to make it suitable for publication.

Good work

Author Response

Dear authors,
the work is interesting, albeit not particularly innovative, as it proposes a medium-term follow-up, however I believe some aspects of it should be improved:

-The name list in the manuscript must be adjusted according to your submission data.

The correct and new autorship change form has been uploaded and sent to the editor

-In the material section authors should specify the setting and how the patient were recruited, the sample selection represents an important source of bias; nothing is specified about the initial characteristics of the WLS of the patients enrolled in the study.

All patients in private practice and presenting WSLs were enrolled, up to 40 lesions. Therefore, no randomisation criteria or WSLs selection criteria were used.

Although among the previously established inclusion criteria there was “non-cavity, unrestored, idiopathic or post-orthodontic white spot lesions (WSLs)”, all patients seen, with the presence of WSLs, were in accordance with the inclusion criteria, none therefore been excluded. None of the lesions included in this research also clashed with the exclusion criterion "dental diseases: hypocalcification, hypoplasia, fluorosis, and hypoplastic molar-incisive syndrome", therefore none of these 40 were excluded.

WSLs were not characterized before being included in the research: their width, depth, or how white they were were not measured. The only criterion is that they were, as explained, "non-cavity, unrestored"

-How was possible to blind the observer if there is just one treatment option evaluated? Considering that to evaluate the effectiveness of the treatment it is necessary to compare the treated tooth with the adjacent one?

The therapy was performed by an operator.

Spectrophotometer measurements were performed by a second operator

The collection of numerical data, the application of the formula to evaluate the color change, and then the collection of the results was performed by a third operator.

The evaluation of the efficacy of the treatment was not carried out by comparing the treated tooth to the adjacent tooth, but by comparing the color of the WSLs to the adjacent sound enamel (SAE) (line 162)

-How was the photographic check mentioned in the study carried out?

The photographic control was carried out using a Nikon d3200 with ring flash, 105mm macro lens and the following shooting parameters: f22, iso 100, white balance 5600k, time 1/125 S, in order to obtain the most realistic photos. However, the photographic check was carried out only for medico-legal purposes (conservation of pre- and post-treatment cases) and for communication with the patient. The analysis of the results was carried out only by spectrophotometer as explained in the article, as an instrument considered the gold standard in color evaluation.

-The results should be reported in more detail, explaining their significance;

The significance of the Wilcoxon test at T0-T1 shows that the technique is effective in determining a color change. The non-significance of the Wilcoxon test at T1-T2 and T1-T3 explains instead that after applying the technique and changing the color of the lesion, there was no color change after 12 and 48 months, which is kept stable.

[This part has been added in the results]

-The discussion must be improved, the first part is redundant.

The discussion has been improved

overall I think the work could be improved to make it suitable for publication.

The English review and implementation was done by a qualified center and the certification was sent to the editor.

Round 2

Reviewer 2 Report

Dear Authors, 
congratulations for the work done.

However, i stil have some minor concerns:

-I still don't agree with the use of the term "blind operator" in your study, it seems to me to be used improperly and only to make the text sound more scientific;

-Discussions start with "However," I would reword the sentence; in my opinion authors should emphasize what emerges from long-term observation, compared to short and medium-term studies already present in the literature, otherwise the study loses significance.

Best regards

Author Response

Dear Authors, 
congratulations for the work done.

Dear Reviewer, Thanks for the compliments

However, i stil have some minor concerns:

-I still don't agree with the use of the term "blind operator" in your study, it seems to me to be used improperly and only to make the text sound more scientific;

Blind operator was removed from the text, we only specified that the second and third operators were unaware of the study

-Discussions start with "However," I would reword the sentence; in my opinion authors should emphasize what emerges from long-term observation, compared to short and medium-term studies already present in the literature, otherwise the study loses significance.

The sentence in the discussion has been reformulated and the term “however” deleted. Thank you
